# Extreme Ultra-Trail Race Induces Muscular Damage, Risk for Acute Kidney Injury and Hyponatremia: A Case Report

**DOI:** 10.3390/ijerph182111323

**Published:** 2021-10-28

**Authors:** Miguel Lecina, Isaac López, Carlos Castellar, Francisco Pradas

**Affiliations:** 1Faculty of Health and Sports Sciences, University of Zaragoza, 22001 Huesca, Spain; miglecina@gmail.com (M.L.); isaac@unizar.es (I.L.); 2ENFYRED Research Group, University of Zaragoza, 22001 Huesca, Spain; franprad@unizar.es

**Keywords:** multi-stage endurance sport, renal impairment, fluid replacement, electrolyte disbalance, muscle damage

## Abstract

A case study involving a healthy trained male athlete who completed a 786 km multi-stage ultra-trail race. Several markers were analyzed in blood and urine samples: creatinine (SCR) for kidney damage, sodium ([Na^+^]) for hyponatremia, creatine kinase (CK) for exertional rhabdomyolysis, as well as other hematological values. Samples were taken before and after the race and during the recovery period (days 2 and 9 after the race). Results showed: SCR = 1.13 mg/dl, [Na^+^] =139 mmol/l and CK = 1.099 UI/l. Criteria for the determination of acute kidney damage were not met, and [Na+] concentration was above 135 mEq/L, indicating the absence of hyponatremia. Exertional rhabdomyolysis was suffered by the athlete (baseline CK increased fivefold), though this situation was reverted after 9 days of recovery. Ultra-trail races cause biochemical changes in athletes, which should be known about by healthcare professionals.

## 1. Introduction

Endurance events have increased their popularity over the last decade [1]. In the USA, 546 races took place in 2016, and this number increased to 1073 in 2019 [2]. One of the most extreme of these sports is ultra-trail races [3]. These events involve considerable efforts, great elevation slopes, both positive and negative, and often severe weather conditions [4], which can develop serious health issues in some runners [5,6,7]. Acute kidney injury (AKI), exertional rhabdomyolysis (ER), and exercise-associated hyponatremia (EAH) are among the effects on runners’ health when research focuses on this sport [8,9].

These races, particularly extra-long ones set in extreme situational conditions, can cause serious health effects on runners, such as the onset of acute kidney injury (AKI), which may or may not be accompanied by EAH. AKI includes structural damage such as a decreased kidney function [10], which could be associated with a significant increase in morbidity and mortality both in the short term and the long [9]. The incidence of AKI episodes found in ultra-trail races is very heterogeneous, varying between 0% [11,12,13,14] and 76% [15]. This wide range of results is due to the different criteria and biomarkers developed to diagnose this illness. The diagnostic of AKI is based on serum creatinine (SCR) or estimated glomerular filtration rate (eGFR) or diuresis [16]. All these markers are compared with baseline, and their increases after completing the race imply acute damage on the kidney of the runners [15,17] and sometimes chronic damage that, ultimately, may even require medical treatment or hospitalization [18].

Contrary to AKI, EAH is easily diagnosed by the concentration of sodium ion [Na^+^] in blood [19], with values under 135 mEq/L set as EAH. Bodyweight is widely recognized as one of the most relevant factors linked to EAH and excessive consumption of beverages with or without [Na^+^] supplementation is the main cause of EAH [20,21]. However, there is still no consensus in the ultra-trail hydration guidelines regarding how much liquid should be consumed during the race, nor much bodyweight athletes should lose during the race [21,22]. By decreasing their bodyweight, runners seem to prevent from suffering EAH. However, neither the exact percentage of weight that runners must lose during the race nor the best hydration protocol for ultra-races has been defined [23].

Another key factor in the development of AKI and/or EAH [24] is post-exertional rhabdomyolysis (ER) [25]. ER is defined as the damage in the muscle characterized by myocellular morphological alterations, and as a result, protein leakage can occur [26,27]. The destruction of the myocyte is often related to any sport, especially those which imply long duration and strenuous intensity because of the mechanical damage and metabolism alterations produced by eccentric exercise [28]. In a systematic review, Rojas et al. [25] studied the cases of ER in different endurance sports, including ultra-trail. Identified cases were classified according to activity type as follows: walking = 1 (0.13%), swimming = 1 (0.1%), spinning = 30 (3.8%), combined activities = 90 (11.4%), cycling = 138 (17.4%), and running = 533 (67.2%). From the total of 130 cases reported with ER + AKI, 96.9% of participants were runners. Another study that compared ultra-trail to other endurance sports was [29], where running and cycling were compared. The cases of EAH were very similar: 6.7% cycling vs. 14.4% in ultra-trail.

The high negative elevation that defines most ultra-trail races offers the opportunity to research the relationship between eccentric exercise and ER. Several studies has proved that some biomarkers related to ER, increase their values during the race and even some days after completing the test [30,31]. The main markers related to ER in scientific reports are Lactodeshidrogenasa (LDH) and creatine kinase (CK) among liver alterations and inflammatory biomarkers such as leucocytes and protein c-reactive [32,33,34]. The diagnosis of ER remains unclear though, with many different values existing to match ER criteria [24].

This case study outlines the main variations in some urine and blood-determined biomarkers relevant to the three aforementioned conditions, as well as hematological and biochemical changes after a 786 km multi-stage ultra-trail race of 11 consecutive stages.

## 2. Case Report

A 42-year-old male athlete with 5 years of experience in ultra-trail races (172 cm, 77.3 kg, 8.14% body fat and 25.6 BMI) took part in this study. The subject is a non-professional runner but with broad experience in ultra-trail races (5 years of expertise). The runner did not present any medical condition or pathology which could interfere with the practice of ultra-trail running. His diet was balanced, and his sleep patterns were totally normal. The subject’s maximum oxygen uptake (VO2max) was 50.71 mL·kg^−1^·min^−1^ and his average weekly training volume was 11 h of running, with 3500 m of cumulative elevation gain. No previous relevant medical history or chronic conditions that limit physical exercise existed. He completed the multi-stage ultra-trail race, which joins the Mediterranean and Atlantic coasts along the Pyrenees, covering 786 km in a total of 11 stages. The race had a warm temperature, with values ranging from 13.08 to 17.69 °C, and the humidity was (60.16–70.87%). It took the athlete 152 h 41″ at an average speed of 5.2 km/h (equivalent to 51% of VO2max). The average stage/day consisted of 71.49 km (SD ± 8.2) and 6457 m (SD ± 663.73) of elevation gain. In-race hydration was ad libitum. Body weight was measured before and immediately after each stage and was recorded both as absolute values and as percentages of body weight loss. Total weight loss was 1.9 kg − 1.8% of body weight. The plasma volume of the subject was 4.830 liters before the race and after completing the race 4690 liters. The volume plasma shift was calculated using the equation revisited of Dill and Costill revisited [35]. Using the following parameters: hemoglobin (pre and post), hematocrit (pre and post), and the plasma volume (pre and post), we estimated that the volume shift that the runner suffered was −1.88%.

Blood and urine samples were taken one day before the race (pre), at the end of the race (post), and on days 2 (rec2) and 9 (rec9) of the recovery periods. Blood samples were collected in two 5 mL Vacutainer tubes (Vacutainer, beliver industrial state, plymouth PL6 7BP, United Kingdom) without anticoagulant for serum isolation and in two 5 mL tubes containing ethylenediaminetetraacetic acid (EDTA) as an anticoagulant. Once collected, blood samples were coagulated for 25–30 min at room temperature and then centrifuged at 2500 rpm for 10 min to remove the clots. Serum samples were aliquoted into Eppendorf tubes (Eppendorf AG, Hamburg, Germany), previously washed with diluted nitric acid, and conserved at −80 °C until the biochemical analysis. For the determination of muscle damage markers and hydration status, a 2 mL blood sample was used. The analyzed biomarkers were serum creatinine (SCR) for AKI, creatinine kinase (CK) for ER, and sodium ion concentration ([Na^+^]) for EAH. Urine sediment analysis was performed with a focus on erythrocytes, leukocytes, and proteinuria. The main hematological and biochemical markers were also analyzed. Results obtained after the GR-11 are displayed as tables. Table 1 shows the blood results and Table 2 the urine results. 

Renal function evaluated through SCR was found to be elevated compared with basal levels from post (+28.41%) through to the 2nd recovery day (rec2 = +11.36%) and dropped below the basal value on recovery day 9 (rec9 = −4.55%). ER measured through CK increased significantly post (+1069.15%) and returned to normal values on rec9. Aspartate aminotransferase (AST) and alanine aminotransferase (ALT), both linked to ER, remained well above baseline levels on recovery day 9 (rec9 AST = +309.52%; rec9 ALT = +607.14%). The electrolyte balance ([Na^+^]), associated with EAH, remained above 135 mEq/L throughout all the recovery phases. This situation did not meet the diagnostic criteria for EAH. Urine sediment analysis did not show evidence of hematuria or proteinuria, despite having found high values of microalbumin post (+18.84%). Values returned to normal during the recovery period (rec2 and rec9). Lastly, hematocrit dropped post (−7.14%) and rec2 (−7.14%), and leukocytes increased by 213.3%, 100%, and 126% for post, rec2, and rec9, respectively.

## 3. Discussion

Renal function assessed through SCR (the main marker for AKI) did not meet the criteria for considering kidney damage since SCR did not increase by 50% [10]. The elevated post-SCR value (+28.41%) returned to normal during recovery and even dropped below basal values. This recovery pattern has been previously reported in other similar studies [36,37] and allows for the determination that nine recovery days are enough for SCR normalization without medical treatment. However, other studies have found several cases of AKI reaching failure stage in one runner (25%) [38]. Due to the seriousness that AKI implies for runners’ health, some studies have addressed the probability of suffering AKI after a previous episode. In this study, 16 runners met the criteria at the first race; the subsequent race caused less increase in SCR concentration and decrement in estimated glomerular filtration rate than the first race. This pattern confirms that usually, runners recover from the increase in SCR after some days [11,13,39,40]

A comparison between multi-stage races and single-stage races shows that multistage races may help runners to recover from the efforts [15,39] reducing the number of episodes diagnosed in these kinds of ultra-trail races; this result may be due to the slower pace that runners use to cover higher distances [40]

One of the reasons behind EAH development in long-distance runners is a poorly planned hydration strategy [41]. Not only does this directly affects EAH, but it has also been described in cases of kidney damage and even ER, due to myoglobin released into the bloodstream following sarcolemma rupture [42,43]. Athlete weight loss should be considered for the evaluation of this aspect, and the current case study results show a weight loss of 1.8% of pre-body weight. This result contrasts with other research that suggests larger weight losses are needed to avoid EAH [44]. This inconsistency regarding EAH diagnosis can be solved by evaluating [Na^+^] concentration, with values above 135 mg/dl positively indicating EAH. In this case, a bodyweight loss below 2% would not be significant for the diagnosis of EAH in such a long duration effort if the [Na^+^] value remains below 135 mEq/L.

ER is the pathological condition involving muscle cell necrosis and the release of CK and myoglobin from muscle cells into tissues, finally carrying to the kidney. There is no general agreement as to exactly how the substances released cause AKI [26,45]. In addition to molecules leaked into blood, vasoconstriction and ischemia seem to be behind the etiology of AKI [10]. Myoglobin and hemoproteins filtered from glomeruli may cause damage to the tubular system affecting the perfusion of the kidney [16,46,47].

As far as the diagnosis of ER is concerned, many blood markers (CK, LDH, AST y ALT) and urine (hematuria, proteinuria) biomarkers have been tested and used to examine ER in sports. All of them have been linked to muscle tissue damage and necrosis [37,48]. However, CK is the most widely used molecule in ER studies [24] and the one which enables its detection and diagnosis [25].

Ultra-trail races imply a huge amount of eccentric muscle contraction because of the great elevation both positive and negative that define these kinds of races [4]. In comparison, the pace and the speed of the runners is slower than in marathon or in half-marathon however the apparition of cases of ER is higher [25]. This could be explained not only by the eccentric effort but also because of hydration issues, which lead to unexperienced runners to EAH and subsequently to ER [29,47].

As a conclusion, the alterations found in this case are similar to other studies, which included races as long and extreme as included in this study [36,46]. Similarly, the subject recovered baseline values after nine days of recovery. The alteration in other makers such as AST or ALT would increase the chances of ER development and also potentially alter liver function, but it is not a criteria for ER [25]. Urine analysis did not show proteinuria or hematuria, which proves that ER is compatible with no visible abnormalities in urine markers [18] (see Table 2).

## 4. Conclusions

This paper shows that ultra-trail races produce alterations in AKI, EAH y ER—specific biomarkers. The number of diagnostic criteria, biomarkers, and the confusing symptomatology often led to these alterations not being adequately considered and regarded as simply adaptation derived from physical effort. Overhydration, intense exercise and eccentric muscle contractions create a perfect storm which can lead to some runners suffering from ER, AKI and EAH. The seriousness of these pathologies, alone or combined, should raise awareness to runners, coaches, and organizers of these races of EAH, AKI, and ER.

## Figures and Tables

**Table 1 ijerph-18-11323-t001:** Blood and urine parameters before (baseline) and after race (post-exercise day 2 and post-exercise day 9).

Parameter Blood	Before-Race	Post-Race		
	Pre (Baseline) Value	Post (Post-Exercise) Value (% Difference)	Day 2 (rec2) Value (% Difference)	Day 9 (rec9) Value (% Difference)
Hemoglobin (g/dL)	144	131 (−9.03)	132 (−8.33)	146 (+1.58)
Hematocrit (%)	42%	39% (−7.14)	39% (−7.14)	43% (+2.38)
RBC (10^6^/mL)	4.42	4.06 (−8.14)	4.05 (−8.37)	4.49 (+1.58)
MCV (fL)	95.6	95.4 (−0.21)	95.6 (0.0)	96.7 (+1.15)
MCH (pg)	32.6	32.4 (−0.61)	32(−1.84)	32.5 (−0.31)
MCHC (g/dL)	341	340 (−0.29)	340 (−0.29)	336 (−1.47)
RDW (%)	12.8	13.5 (+5.47)	13.7 (+7.03)	13.8 (+7.81)
Platelet count (10^6^/mL)	242	315 (+30.17)	309 (+27.69)	391 (+61.57)
Platelet volume (fL)	8.1	7.6 (−6.17)	7.7 (−4.94)	7.3 (−9.88)
Leukocytes (10³/mL)	3	9.4 (+213.33)	6 (+100.00)	6.8 (+126.00)
Neutrophils (10³/mL)	3	6.5 (+116.67)	3.6 (+20.00)	3.7 (+23.33)
Neutrophils (%)	52.2	68.7 (+31.61)	60.9 (+16.67)	53.7 (+2.87)
Lymphocytes (10³/mL)	2.1	1.7 (−19.05)	1.4 (−33.33)	2.3 (+9.52)
Lymphocytes (%)	35.3	17.8 (−49.58)	23.5 (−33.43)	34.5 (−2.27)
Monocytes (10³/mL)	0.6	1 (+66.67)	0.5 (+16.67)	0.6 (0.00)
Monocytes (%)	10	10.4 (+4.0)	9 (−10.00)	8.6 (−14.00)
Eosinophils (10³/mL)	1.8	0.2 (−89.89)	0.3 (−83.34)	0.1 (−94.44)
Eosinophils (%)	1.8	2.4 (+33.33)	5.3 (+194.44)	2.1 (+16.67)
Basophils (10³/mL)	0.0	0 (+0.10)	0 (+0.10)	0 (+0.10)
Basophils (%)	0.7	0.7 (0.00)	1.3 (+85.71)	1.1 (+57.14)
Erythroblasts (10³/mL)	0.0	0 (0.00)	0 (0.00)	0 (0.00)
Erythroblasts (%)	0.0	0 (0.00)	0 (0.00)	0 (0.00)
SCR (mg/dl)	0.88	1.13 (+28.41)	0.98 (+11.36)	0.84 (−4.55)
AST (UI/L)	21	66 (+214.29)	45 (+114.29)	86 (+309.52)
ALT (UI/L)	14	39 (+178.57)	33 (+135.71)	99 (+607.14)
Lipase (UI/L)	13	18 (+38.46)	27 (+107.69)	13 (0.0)
Urea (mg/dL)	33	64 (+93.94)	46 (+39.39)	35 (+6.06)
Uric Acid(mg/dL)	5.2	5 (−3.85)	4.7 (−9.62)	5.2 (0.00)
HDL Cholesterol (mg/dL)	90	86 (−4.44)	80 (−11.11)	94 (+4.44)
Total Cholesterol (mg/dL)	233	193 (−17.17)	194 (−16.74)	292 (+25.32)
Triglycerides (mg/dL)	79	80 (+1.27)	130 (+64.56)	88 (+11.39)
Na^+^ (mmol/L)	136	139 (+2.21)	140 (+2.94)	137 (+0.74)
K^+^ (mmol/L)	4.5	5.2 (+15.56)	5.5 (+22.22)	4.9 (+8.89)
Cl^+^ (mmol/L)	102	107 (+4.90)	106 (+3.92)	99 (−2.94)
Ca^2+^ (mg/dL)	9.9	9.1 (−8.08)	9.1 (−8.08)	9.8 (−1.01)
Mg^2+^ (mg/dL)	2	2.1 (+5.00)	2 (0.00)	2.2 (+10.00)
P^+^ (mg/dL)	2.9	3.5 (+20.69)	2.9 (0.00)	3.4 (+17.24)
Glucose (mg/dL)	92	99 (+7.61)	73 (−20.65)	95 (+3.26)
Albumin (g/dL)	4.3	3.89 (9.53)	3.64 (−14.35)	4.29 (−0.23)
CK (UI/L)	94	1099 (+1069.15)	478 (408.51)	109 (+15.96)
LDH (UI/L)	152	571 (+275.66)	422 (+177.63)	254 (+67.11)
Total Proteins (g/dL)	6.8	6.5 (4.41)	6.1 (−10.29)	7 (+2.94)
Urea (mg/dL)	33	64 (+93.94)	46 (+39.39)	35 (+6.06)

Data are expressed as absolute value and as ± percentage from baseline values AST, aspartateaminotransferase; ALT, alanineaminotransferase; CK, creatine kinase; HDL, high-density lipoprotein; K^+^, ion potassium LDL, low-density lipoprotein; LDH, lactate dehydrogenase; MCH, mean corpuscular hemoglobin; MCHC, mean corpuscular hemoglobin concentration; MCV, mean corpuscular volume; RBC, red blood cell; RDW, red blood cell distribution width; SCR, creatinine.

**Table 2 ijerph-18-11323-t002:** Urine Parameters Before (Baseline) and After Race (Post-Exercise Day 2 and post-exercise Day 9).

Parameter Urine	Before-Race	Post-Race		
	Pre (Baseline) VALUE	Post (Post-Exercise) Value (% Difference)	Day 2 (rec2) Value (% Difference)	Day 9 (rec9) Value (% Difference)
Proteins (mg/dL)	0	0 (0.00%)	0 (0.00%)	0 (0.00%)
Density (Kg/L)	1021	1021 (0.00%)	1018 (−0.29%)	1014 (−0.69%)
PH	6.5	6 (−7.69%)	7 (+7.69%)	7 (+7.69%)
Glucose (mg/dL)	0	0 (0.00%)	0 (0.00%)	0 (0.00%)
Nitrites	0	0 (0.00%)	0 (0.00%)	0 (0.00%)
Ketonic Bodies (mg/dL)	0	0 (0.00%)	0 (0.00%)	0 (0.00%)
Leucocytes	0	0 (0.00%)	0 (0.00%)	0 (0.00%)
Erythrocytes	0	0 (0.00%)	0 (0.00%)	0 (0.00%)
Microalbumin (mg/dL)	<0.19	36 (+18,847%)	<0.19 (0.00%)	<0.19 (0%)
Bilirubin (mg/dL)	0	0 (0.00%)	0 (0.00%)	0 (0.00%)
Urobilinogen (mg/dL)	1	1 (0.00%)	1 (0.00%)	0.2 (−80.00%)

Data are expressed as absolute value and as +. − percentage from baseline value.

## Data Availability

Information about the case report is available at http://gr11en11.org/ (accessed on 26 Oct 2021).

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
