# Peer review of "Extreme Ultra-Trail Race Induces Muscular Damage, Risk for Acute Kidney Injury and Hyponatremia: A Case Report"

_ijerph, 2021, doi:10.3390/ijerph182111323_

Round 1

Reviewer 1 Report

Ultra-trail racing involves extreme efforts, but the research approach about
this type of event can never be based on the results obtained from a case.

The variations in urine samples and biomarkers determined in blood are not
relevant, if they only refer to an individual, despite being subjected to
extreme conditions.

What is relevant for the three scientific knowledge of response to such
extreme conditions, is to confirm that these changes are manifested in
different individuals who undergo the same tests.

The study of a case is only interesting in biomedicine for rare diseases. These physical tests, being extreme, are performed by a substantial number of individuals,who deserve to be included in the results and not a single athlete.

Furthermore, the biochemical parameters analyzed do not provide relevant
or novel information, they are analytes that are traditionally used and do
not provide new scientific information.

Author Response

Dear reviewer,

Thank you for giving us the opportunity to submit a revised draft of our manuscript titled “Extreme Ultra-Trail Race Induces Muscular Damage, Risk for Acute Kidney Injury and Hyponatremia: A case report.” to "International Journal of Environmental Research and Public Health".

We assume that a larger number of subjects would have increased the effect size of our study. In the same way, the information about the medical conditions investigated in our case study.

We accept that a larger data of population would have increased the effect size of our study as well as the information about the medical conditions investigated in our case study. However, due to the extreme conditions of the race included in our case study and the difficulty of the challenge, we decided to use a case study to acquire data preserving the integrity and health of the runner.  

In medicine, case studies may be restricted only for infrequent illnesses or medical conditions uncommon, but in Sport or Physical Activity Science, this design has been carried out several times. Especially in extreme sports like ultra-trail. The length, the singularity of the race and the elevation of this race, deserves in our humble opinion a case study

We hope you may find convenient the information added in this email, and please do not hesitate to contact us regarding any queries you might have.

Yours faithfully,

Reviewer 2 Report

Please include my notes in your contribution.

Author Response

Dear reviewer,

Thank you for allowing us to submit a revised draft of our manuscript titled “Extreme Ultra-Trail Race Induces Muscular Damage, Risk for Acute Kidney Injury and Hyponatremia: A case report.” to "International Journal of Environmental Research and Public Health".

We appreciate the time and effort that you have dedicated to providing your valuable feedback on our manuscript. Consequently, we have been able to incorporate changes to reflect most of the comments provided by you. We have highlighted (yellow color) the changes within the manuscript (Please see the attachment).

Here is a point-by-point response to your main notes and concerns. Additionally following the manuscript that you gently attached in the review process we have corrected minor issues as misspelling or format corrections.

  • Comment 1

“a grammatical editing of the entire paper is required”

  • Response 1: Agree. We have edited the whole document and corrected grammar mistakes and misspellings to solve this point.
  • Comment 2

In the introduction, it would be appropriate to mention other related studies to get a better understanding of the related issues.

  • Response 2: We have added several references (24) and have described the incidence and aetiology of the three medical conditions studied in this paper. In total, we have added 642 words, and four paragraphs focusing on other studies similar to our case study offering relevant information to fully understand the AKI, EAH and ER issue in ultra trail races and endurance sports in general.
  • Comment 3

“Discussion should be more in-depth.”

  • Response 3: We have added 681 new words and 30 references. Plus, we have dedicated a whole paragraph to every one of the medical conditions referred to in the case study.

The association between them has been studied, as well as the circumstances that foster the apparition of them. The different characteristics of the ultra-trail races have been researched, including (length, number of stages, elevation positive and negative). Plus, the aetiology and possible physiopathology of AKI, EAH and ER have been described showing the different theories existing nowadays. Additionally, the aetiology and possible physiopathology of AKI, EAH and ER have been theorized, discussing the current explanations for these pathologies.

We hope you may find convenient the information added in this email, and please do not hesitate to contact us regarding any queries you might have.

Yours faithfully,

Reviewer 3 Report

  • Author should explain additional pathophysiology on Acute Kidney Injury among Ultra-trail Races. P. 28
  • Details on blood biochemistry (serum, plasma), including techniques to look at the blood samples, why looking at those markers only? P. 47
  • Discussion need to elaborate more with explanations in different endurance activity.

Author Response

Dear reviewer,

Thank you for giving us the opportunity to submit a revised draft of our manuscript titled “Extreme Ultra-Trail Race Induces Muscular Damage, Risk for Acute Kidney Injury and Hyponatremia: A case report.” to "International Journal of Environmental Research and Public Health".

We appreciate the time and effort that you have dedicated to providing your valuable feedback on our manuscript. In consequence, we have been able to incorporate changes to reflect most of the comments provided by you. We have also highlighted the changes within the manuscript (green color) to facilitate the location of the changes (please see the attachment).

Here is a point-by-point response to your main notes and concerns. Additionally following the manuscript that you gently attached in the review process we have corrected minor issues as misspelling or format corrections.

  • Comment 1

Author should explain additional pathophysiology on Acute Kidney Injury among Ultra-trail Races. P. 28

  • Response 1: Agree. We have added a physiological explanation for Acute kidney damage (P.35 – P.40) as well as for Hyponatremy (P. 56 - P.68) and Exertional Rhabdomyolysis (P. 62 to P.66). These modifications have been highlighted in the manuscript to ensure you can find the corrections in the whole document.
  • Comment 2

Details on blood biochemistry (serum, plasma), including techniques to look at the blood samples, why looking at those markers only? P. 47

  • Response 2: We have added the procedures and methodology followed in the studio to collect, process and finally analyze the blood and the urine samples (P. 96 to P. 111). Several additional biomarkers including, red blood cells, white blood cells and biochemical markers have been analyzed and are shown in detail in table 1 and table 2
  • Comment 3

“Discussion need to elaborate more with explanations in different endurance activity”

  • Response 3: We have added seven new references and one paragraph to compare ultra-trail and other endurance sports as you suggested. (P.145 to P. 156). Plus, we have dedicated a whole paragraph to every one of the medical conditions referred to in the case study. The association between them has been studied, as well as the circumstances that foster the apparition of them. The different characteristics of the ultra-trail races have been researched, including (length, number of stages, elevation positive and negative). Additionally, the aetiology and possible physiopathology of AKI, EAH and ER have been discussed above, showing the different possible explanations proposed nowadays. In addition, the aetiology and possible physiopathology of AKI, EAH and ER have been theorized, showing the current theories present in scientific literature.

We hope you may find convenient the information added in this email, and please do not hesitate to contact us regarding any queries you might have.

Yours faithfully,

Round 2

Reviewer 1 Report

I do not consider the work to be suitable for scientific publication.
The results do not provide any new information on previous knowledge nor are they transferable for practical application. The study of a case is acceptable to be published in the medical scientific field, where the appearance of more cases cannot be forced, but it is not acceptable in the sports scientific field, since in the latter, more cases of the same type can be reported of physical activity to be analyzed in the study. The difficulty of incorporating more cases to the study does not justify their not being incorporated, since they exist.

Author Response

Dear Reviewer,

I have read your second comment regarding the convenience of our case study to this special issue of the journal. Regrettably, we can´t share your point because case studies are not limited to the medical field. Ultra-endurance sports are nowadays a cause of concern because of their popularity. The risks that these races imply for the participants´ health justify the need to use a case study. In our opinion, case study is a suitable tool for this approach avoiding unnecessary risks. The case studies involving extreme sports are accepted and included in many JCR journals as this journal. The difficulty, the duration and the positive and negative elevation of this race, in addition to simultaneous medical conditions assessed, deserve to be published as the other reviewers decided. In conclusion, we appreciate your comments and, to the extent possible, will be considered for the new studies our research group is planning.

Yours thankfully